# Mixing state and effective density of aerosol particles during the Beijing 2022 Olympic Winter Games

Aodong Du[1,2], Jiaxing Sun[1,2], Hang Liu[1], Weiqi Xu[1], Wei Zhou[1], Yuting Zhang[1,2], Lei Li[3], Xubing Du[3,4], Yan Li[1,2], Xiaole Pan[1], Zifa Wang[1,2], and Yele Sun[1,2]

[1]State Key Laboratory of Atmospheric Boundary Layer Physics and Atmospheric Chemistry, Institute of Atmospheric Physics, Chinese Academy of Sciences, Beijing 100029, China

[2]College of Earth and Planetary Sciences, University of Chinese Academy of Sciences, Beijing 100049, China

[3]Institute of Mass Spectrometry and Atmospheric Environment, Jinan University, Guangzhou 510632, China

[4]Guangdong Provincial Engineering Research Center for On-Line Source Apportionment System of Air Pollution, Guangzhou 510632, China

*Correspondence to*: Yele Sun (sunyele@mail.iap.ac.cn)

**Abstract.** Mixing state and density are two key parameters of aerosol particles affecting their impacts on radiative forcing and human health. Here a single particle aerosol mass spectrometer in tandem with a differential mobility analyzer and an aerodynamic aerosol classifier was deployed during the Beijing 2022 Olympic Winter Games (OWG) to investigate the impacts of emission controls on particle mixing state and density. Nearly 760,000 particles were detected, which were classified into seven major classes. Our results show the dominance of carbonaceous particles comprising mainly Total-Elemental Carbon (Total-EC, 13.4 %), Total-Organic Carbon (Total-OC, 10.5 %) and Total-ECOC (47.1 %). Particularly, the particles containing organic carbon and sulfate were enhanced significantly during OWG although those from primary emissions decreased. The composition of carbonaceous particles also changed significantly which was characterized by the decreases in EC-Nitrate and Sulfate (EC-NS), EC-Potassium Nitrate (KEC-N), and amine-containing particles, and increase in ECOC-Nitrate and Sulfate (ECOC-NS). This result indicates that emission controls during OWG reduced the mixing of EC with inorganic aerosol species and amines, yet increased the mixing of EC with organic aerosol. The average effective density ($\rho_{eff}$) of aerosol particles (150–300 nm) was 1.15 g cm$^{-3}$ during the non-Olympic Winter Games (nOWG), with higher values during OWG (1.26 g cm$^{-3}$) due to the increase in secondary particle contribution. The two types of fresh particles, i.e., Total-EC and high molecular weight organic matter presented the lowest $\rho_{eff}$ (0.97 g cm$^{-3}$ and 0.87 g cm$^{-3}$, respectively). In addition, the $\rho_{eff}$ of most particles increased as the increases in pollution levels and relative humidity, yet varied differently for different types of particles, highlighting the impacts of aging and formation processes on the changes of particle density and mixing state.

## 1 Introduction

Atmospheric aerosols from both natural and anthropogenic sources exert strong influences on radiative forcing and human health (Buseck and Posfai, 1999; Anderson et al., 2003; Ramanathan et al., 2001; Prather, 2009; Charron et al., 2007), and the impacts depend strongly on their chemical and physical properties, e.g., composition, mixing state, and density. Beijing,

experiencing severe pollution with high concentrations of fine particulate matter over the past decade (Huang et al., 2014; Guo et al., 2014; Sun et al., 2016), has a great success in air pollution control (Lei et al., 2021a; Cheng et al., 2019), and the annual average concentration of PM$_{2.5}$ reached the Chinese National Ambient Air Quality Standard for the first time in 2021 (33 µg m$^{-3}$). However, polluted events still occurred occasionally, particularly during wintertime with stagnant meteorological conditions and high anthropogenic emissions (Lei et al., 2021b; Xu et al., 2022; Feng et al., 2022; Zhou et al., 2023). Short-term emission controls can improve air quality by reducing anthropogenic emissions temporarily, for example, the Beijing 2008 Summer Olympics (Wang et al., 2010; Okuda et al., 2011; Zhou et al., 2010), the 2014 Asia-Pacific Economic Cooperation (APEC) summit (Han et al., 2015; Gao et al., 2017; Ren et al., 2018), and the 2020 lockdown due to the corona virus disease (COVID-19) (Sun et al., 2020; Chang et al., 2022; Rajesh and Ramachandran, 2022; Zhang et al., 2022a). Similarly, the Chinese government imposed strict emission controls in Beijing and surrounding regions during the Olympic Winter Games (OWG) in Beijing in 2022. These measures included shutting down factories with high emissions, limiting the number of vehicles, stopping construction activities, and forbidding fireworks, etc. (The People's Government of Beijing Municipality, 2022). As a result, the PM$_{2.5}$ decreased significantly during OWG (Liu et al., 2022). Although the effects of emission controls on chemical composition and formation mechanisms of fine particles have been extensively studied, their impacts on chemical and physical properties of single particles, e.g., mixing state and effective density ($\rho_{eff}$) are poorly understood.

Single particle aerosol mass spectrometers (SPAMS) have been widely used to characterize mixing state, chemical compositions and formation mechanisms of aerosol particles (Bhave et al., 2002; Giorio et al., 2015; Li et al., 2014; Bi et al., 2011; Chen et al., 2017; Zhang et al., 2018). Previous single particle studies in China were mainly conducted in polluted winters. Chen et al. (2020) found that carbonaceous particles associated with coal combustion were the main contributors of fine particles during wintertime in urban and rural areas of Beijing. Zhang et al. (2022b) found that the proportion of elemental carbon particles mixed with organics gradually increased in winter from 2016 to 2019 in Chengdu. Similarly, elemental carbon particles tended to be more mixed with organics in winter in Beijing compared to summer (Xie et al., 2020). However, the measurements of critical parameters of aerosol particles, e.g., effective density ($\rho_{eff}$) with different mixing state, are still limited (Buseck and Posfai, 1999; Pitz et al., 2003).

$\rho_{eff}$ reflecting the average density and morphological characteristics of particles depends strongly on mixing state, sources and aging levels (Decarlo et al., 2004). To obtain $\rho_{eff}$, SPAMS is often deployed along with other instruments that can measure particle sizes as well. Spencer et al. (2007) deployed a differential mobility analyzer (DMA) and a SPAMS simultaneously, and determined the $\rho_{eff}$ of aerosol particles in the range of ~1 to 1.5 g cm$^{-3}$ in California in summer. Using the same system, Zhai et al. (2017) found that the $\rho_{eff}$ of biomass combustion particles ranged from ~1.16 to 1.51 g cm$^{-3}$. Recently, an aerodynamic aerosol classifier (AAC) tandem SPAMS system was established by Peng et al. (2021) to determine the volume equivalent diameter ($D_{ve}$) and $\rho_{eff}$ of different types of aerosol particles. This system was then used to analyze the characteristics of particles emitted from diesel vehicle engines with different mixing state (Su et al., 2021). The results showed largely different $\rho_{eff}$ for aerosol particles emitted under launching and idling conditions (0.66 vs. 0.34 g cm$^{-3}$). Till now, few studies have characterized the $\rho_{eff}$ of ambient aerosol particles at different mixing state using similar tandem systems, and the evaluation of the impacts of emission controls is rare as well.

In this study, a high-resolution SPAMS coupled with a DMA and an AAC, respectively, was deployed during OWG in Beijing to characterize the chemical and physical properties of aerosol particles. The chemical composition, mixing state, and $\rho_{eff}$ are determined, and the differences between OWG and non-Olympic Winter Games (nOWG) are explored. Particularly, the influences of emission controls on fine particle characteristics and mixing state are elucidated.

## 2 Methods

### 2.1 Sampling site and measurements

All instruments were deployed at Institute of Atmospheric Physics (IAP), Chinese Academy of Sciences (39°58'28"N, 116°22'16"E), an urban site influenced by both residential and traffic emissions. To measure aerosol particles according to density, two tandem systems by coupling differential mobility analyzer (DMA, model 3085A, TSI Inc.) and single particle aerosol mass spectrometers (SPAMS, Hexin Analytical Instrument Co., Ltd.), aerodynamic aerosol classifier (AAC, Cambustion Ltd.) and SPAMS were operated from 21 January to 10 February, and from 10 February to 1 March, respectively (Fig. 1). Aerosol particles were first filtered with a $PM_{2.5}$ cyclone placed in front of the sampling line. After dried with a diffusion dryer, the particles with mobility diameters ($D_m$) of 150, 200, 250 and 300 nm were selected by DMA, and those with aerodynamic diameter ($D_a$) of 300 nm were selected by AAC, and then the highly monodispersed particles were measured by SPAMS. The detailed description of AAC is given elsewhere (Liu et al., 2020; Liu et al., 2019; Yu et al., 2022). Before the campaign, the size calibration of SPAMS was performed using polystyrene latex spheres (PSLs) with known sizes (0.23, 0.32, 0.51, 0.74, 0.96, 1.4, and 2 μm).

A seven-wavelength (370, 470, 520, 590, 660, 880, and 950 nm) Aethalometer (AE33, Magee Scientific Corp.) and a high-resolution time-of-flight aerosol mass spectrometer (HR-ToF-AMS) were deployed at the same site to measure equivalent black carbon (eBC) and non-refractory organics (Org), sulfate ($SO_4$), nitrate ($NO_3$), ammonium ($NH_4$), and chloride (Chl) in $PM_1$. The time resolution of both instruments was 1 min, where HR-ToF-AMS was measured in V-mode and the mass concentration of eBC obtained by AE33 was calculated based on the dual-spot measurement (Rajesh and Ramachandran, 2018; Drinovec et al., 2015). The HR-ToF-AMS data were analyzed by using PIKA v 1.24, which showed that $NO_3$ (4.30 μg m$^{-3}$) and Org (3.80 μg m$^{-3}$) contributed 68.0 % of the mass concentration of non-refractory submicron aerosols (NR-PM$_1$, 11.92 μg m$^{-3}$), followed by $SO_4$ (1.91 μg m$^{-3}$), $NH_4$ (1.69 μg m$^{-3}$), and Chl (0.22 μg m$^{-3}$). Organic aerosols (OA) were analyzed by positive matrix factorization (PMF), and five OA factors were identified including biomass burning and fossil fuel combustion-related OA (FFBBOA), cooking OA (COA), and three SOA factors, i.e., two oxygenated OA (OOA1 and OOA2) and an aqueous-phase OOA with mass concentrations of 0.31, 0.87, 0.83,1.18 and 0.56 μg m$^{-3}$, respectively. More details on the operations and data analysis of AE33 and HR-ToF-AMS are given in Xu et al. (in preparation).

### 2.2 Data analysis

#### 2.2.1 SPAMS

A total of 2619,193 particles were sized and approximately 760,000 particles with both mass spectra and size information were captured by SPAMS (Table S1). As shown in Fig. S1, the counts of total particles correlated reasonably well with non-

refractory submicron aerosols (NR-PM$_1$, $r = 0.76$), and those of EC-containing particles correlated well with eBC ($r = 0.87$). The data were then analyzed using a neural network algorithm based on adaptive resonance theory (ART-2a) (Song et al., 1999). The vigilance factor, learning rate and iterations for ART-2a were set as 0.75, 0.05 and 20, respectively in this study.

According to the mass spectral characteristics, temporal trends and size distributions (Dall'osto and Harrison, 2006; Phares et al., 2001), more than 99 % of the total particles were grouped into seven major classes including Total-EC, Total-ECOC, Total-OC, Total-IA, Biomass-K, HOM and Metals, and several subclasses (Table 1). The classification of particles is based on three principles: (1) particles with distinct $C_n^{\pm}$ (n = 1, 2, 3…) signals are named as EC; (2) particles with distinct OC signals (including $27[C_2H_3]^+$, $37[C_3H]^+$, $43[C_2H_3O]^+$, $50[C_4H_2]^+$, $51[C_4H_3]^+$…) are named as OC; and (3) particles are named

as ECOC when having comparable EC and OC signals in the positive spectra (Sun et al., 2022a; Sun et al., 2022b). The detailed names of particle types and characteristics are given in Table S2, and the average mass spectra of each type of particle are depicted in Figs. 2 and S2.

**2.3.2 Effective density**

Given that DMA and AAC are selected for mobility diameter ($D_m$) and aerodynamic diameter ($D_a$) of particles, respectively,

two methods are adopted to calculate the $\rho_{eff}$ in this study. For the DMA-SPAMS tandem system, the $\rho_{eff}$ is determined as the ratio of vacuum aerodynamic diameter ($D_{va}$) to $D_m$ as Eq. (1).

$$\rho_{eff} = \frac{D_{va}}{D_m} \rho_0 \tag{1}$$

Where $\rho_0$ is the standard density (1.0 g cm$^{-3}$).

For the AAC-SPAMS tandem system, the $\rho_{eff}$ is calculated as the ratio of particle density ($\rho_p$) and particle dynamic shape

factor ($\chi_\gamma$):

$$\rho_{eff} = \frac{\rho_p}{\chi_\gamma} = \frac{D_{va}}{D_{ve}\rho_0} \tag{2}$$

where $D_{ve}$ represents the volume equivalent diameter. A more detailed description of the relationship between $D_a$, $D_{va}$ and $D_{ve}$ is given in Decarlo et al. (2004). Combining the calculated $\rho_{eff}$ with the particle mixing states measured by SPAMS makes it possible to analyze the $\rho_{eff}$ of particles with different compositions.

**3 Results and discussion**

**3.1 Mixing state of aerosol particles**

Figure 3 depicts the time series of meteorological parameters, pollutant concentrations and the number fractions of seven types of particles during the entire study. The average (±1σ) mass concentrations of NR-PM$_1$ and eBC for the entire study were 11.92 (± 15.77) and 1.34 (± 1.41) µg m$^{-3}$, respectively, and they decreased by 48.7 % and 37.5 % during OWG compared

with the nOWG period (Table 2). In addition to emission controls, the favorable meteorological conditions during OWG as indicated by lower RH and $T$, and higher WS also played a role for the low concentrations (Liu et al., 2022).

Figure S3 shows the digital mass spectra of all particles for the whole period. Significant ion peaks of organic and carbon clusters in positive spectra, and nitrate ($46[NO_2]^-$ and $62[NO_3]^-$) and sulfate ($97[HSO_4]^-$) in negative spectra were observed. This was consistent with the AMS measurements that nitrate and organics were the major components of NR-PM$_1$, on average accounting for 36.1 % and 31.9 %, respectively, followed by sulfate (16.0 %). The differences in the relative peak areas of aerosol particles between OWG and nOWG periods were also observed (Fig. S3b). The signals of $39[K]^+$ and organic carbon ($27[C_2H_3]^+$, $43[C_2H_3O]^+$, $50[C_4H_2]^+$) and $97[HSO_4]^-$ were significantly enhanced during OWG, suggesting the increased importance of organic aerosol and sulfate. In addition, we found that the ratio of peak area of sulfate to nitrate ($PA_{sulfate}/PA_{nitrate}$) during OWG (0.26) was slightly higher than that during nOWG (0.24), suggesting the elevated aging of aerosol particles. In fact, the daily captured particle counts from SPAMS show that aging particles become more prominent during the OWG period due to the combination of the significant reduction in the number concentration of primary particles (e.g., Total-EC decreased by 61.80%) and the marked increase in the number concentration of some aging particles, such as ECOC-NS (17.34% increase), during the OWG period (Table S3).

The Total-ECOC particles with intense ion peaks of elemental carbon (EC, $C_n^{\pm}$, n = 1, 2, 3...) and organic carbon (OC, $27[C_2H_3]^+$, $37[C_3H]^+$, $43[C_2H_3O]^+$, $50[C_4H_2]^+$, $51[C_4H_3]^+$...) represented 47.1 % of the total particles, followed by Total-EC (13.4 %) and Biomass-K (13.0 %). The carbonaceous particles accounting for 71.0 % of the total were divided into 12 subclasses (Table 1). These carbonaceous particles were overall mixed with nitrate (relative peak area ~ 0.4) or sulfate (~0.05−0.2), and showed pronounced $39[K]^+$, $23[Na]^+$ or $18[NH_4]^+$ signals in the positive spectra. We observed a clear decrease in Total-EC from 11.5 % to 8.5 % during OWG, while increased contributions for Total-ECOC and Biomass-K by 3.3 and 2.2 %, respectively. These results indicate the changes of mixing state of aerosol particles during OWG. Most importantly, the composition of carbonaceous particles also changed significantly. The largest decreases were observed for EC-NS and KEC-N by 5.5 and 3.4 %, respectively, while the proportion of ECOC-NS increased significantly from 16.8 % to 28.4 %. In addition, the ammonium-containing ($18[NH_4]^+$) and trimethylamine-containing ($58[C_3H_8N]^+$ and $59[C_3H_9N]^+$) particles were largely reduced from 8.7 % to 1.1 % during OWG. Such results indicate that emission controls during OWG reduced the mixing of EC with inorganic aerosol species and amines, yet increased the mixing of EC with organic aerosol.

The Biomass-K particles were identified with intense $39[K]^+$ and levoglucosan ion peaks ($45[CHO_2]^-$, $59[C_2H_3O_2]^-$, $71[C_3H_3O]^-$ and $73[C_3H_5O_2]^-$). Pratt et al. (2011) found that levoglucosan can degrade rapidly due to atmospheric oxidation. Therefore, the contribution of 13.0 % in this measurement indicated that the Biomass-K particles may undergo atmospheric oxidation processes to some extent. A class of high-molecular-weight organic matter (HOM) characterized by distinct polycyclic aromatic hydrocarbons (PAHs) e.g., $152[C_{12}H_8]^+$, $165[C_{13}H_9]^+$, $178[C_{14}H_{10}]^+$, and $189[C_{15}H_9]^+$ was also detected (Zhang et al., 2022c), accounting for 5.5 % of the total particles. The prominent nitrate ($46[NO_2]^-$ and $62[NO_3]^-$) signals in negative spectra indicated the mix of HOM particles with secondary inorganic aerosol species (Fig. 2e). In addition, SPAMS also detected some relatively pure inorganic aerosol (IA) particles that did not mix with other components, and metal particles ($55[Mn]^+$, $56[Fe]^+$ and $206,207,208[Pb]^+$) that were likely from anthropogenic emission and road dust. A large number of rich-Fe particles with peak area ratio $56[Fe]^+/54[Fe]^+ > 3$, accounting for 59.1 % of Metals particles, were mainly observed during periods with snowfall events, and were strongly associated with secondary species especially nitrate (Fig. S2). Total-IA and Metals accounted for 7.0 % and 3.5 % of the total particles, respectively, both of which showed decreased contributions

during OWG, indicating the suppressed local secondary formation due to reduced precursors and the effect of stopping construction activities.

## 3.2 Diurnal cycles and sources

Figure 5 depicts the diurnal variations of the normalized counts of different types of particles throughout the campaign. Particles including KECOC-NS and KOC-N showed similar diurnal cycles with the lowest values occurring at 16:00 and high values at night. The high correlations of these particles with primary OA from fossil fuel combustion and biomass burning ($r$ = 0.78 and 0.70, Table S4) highlight their dominant sources of primary emissions. Consistently, the dominant contribution (OWG vs. nOWG: 48.1 vs. 47.2 %) of KECOC-NS and KOC-N to the carbonaceous particles at low $PA_{sulfate}/PA_{nitrate}$ (< 0.2), an indicator of particle aging level (Li et al., 2020), supported the properties of fresh emissions as well (Fig. 6). The bivariate polar plots (Fig. S4) indicated that these two types of particles were mainly transported from the southeast during OWG. This result suggests that fresh particles in winter in Beijing could also be from regional transport over a small scale under emission control. Previous studies have demonstrated that particles with significant organic nitrogen fragments ($26[CN]^-$ and $42[CNO]^-$) and $39[K]^+$ signals may come from wildfires, biomass burning and coal combustion (Pratt et al., 2011; Zauscher et al., 2013; Hu et al., 2021b; Bi et al., 2011). Considering the strict emission controls in Beijing during OWG, the higher normalized count of KECOC-CN at night was likely attributed to the regional transport of primarily emitted particles near Beijing (Fig. S4). However, the low peak area ratio of EC to OC ($PA_{EC}/PA_{OC}$) for ECOC-containing particles indicated overall higher aging levels during OWG (Fig. 7e-j) (Pio et al., 2011; Pokhrel et al., 2016). Particularly, the $PA_{EC}/PA_{OC}$ of KECOC-NS and KECOC-CN decreased obviously from 1.01 to 0.82, and 0.92 to 0.80, respectively, during OWG.

The pronounced diurnal cycle of HOM indicated the sources of coal combustion and traffic emissions from heavy duty vehicles and diesel trucks, and the transport from the southeast (Fig. S4), which reached a maximum weight in total particles around 8:00 (11.4 % for OWG and 10.2 % for nOWG, Fig. 4). A pronounced diurnal cycle with high values in the morning (~11:00) was also observed for ECOC-NS, likely indicating the similar sources as HOM, yet the strong $97[HSO_4]^-$ signal in mass spectra suggested more aged properties. The moderate correlation ($r$ = 0.63) between ECOC-NS and HOM and the bivariate polar plots especially during nOWG (Fig. S4b) also supported this conclusion. In fact, the ECOC-NS particles are important across different $PA_{sulfate}/PA_{nitrate}$ values demonstrating the complexity of its sources (Fig. 6b, d).

In addition, KNa-containing carbonaceous particles are generally considered to be from the incomplete combustion of solid fuels such as coal combustion and traffic emissions (Xie et al., 2020; Hu et al., 2021b; Li et al., 2018). KNaEC-N and KNaECOC-NS particles were mixed a considerable nitrate (Fig. S2). The high correlations with FFBBOA and chloride (Table S4), and the bivariate polar plots of the particles emphasized the features of local emissions.

The types of KEC-N, EC-NS, KAECOC-NS, K-Amine-NS and K-N particles were closely associated with three SOA factors (Table S4). KEC-N and EC-NS accounted for ~75.4 % of Total-EC with relatively small daily variations (Fig. 5f) and greater contribution in the afternoon, which was particularly evident during nOWG (Fig. 4d, h). This result might indicate the background or regional characteristics of these particles, which was consistent with the conclusion of Dall'osto et al.

(2016). Interestingly, the contribution of Total-EC to the total particles decreased from 15.4 % to 5.5 % as the increase of $PA_{sulfate}/PA_{nitrate}$ from 0 to 1.1 during nOWG (Fig. 6c), whereas the change of Total-EC contribution was relatively flat (from 10.3 % to 7.0 %) during OWG (Fig. 6a). This was mainly attributed to the emission control which led to a decrease in the proportion of EC-containing particles at low aging levels. The minimum daily values of both KAECOC-NS and K-N particles appeared at 8:00 and gradually increased thereafter (Fig. 5i, l). Chen et al. (2020) suggested that the increase in K-N was mainly achieved through the uptake of nitrate from daytime photochemical production. KAECOC-NS particles characterized by high ammonium ($18[NH_4]^+$) signal were mainly observed during polluted periods with snowfall. Similarly, K-Amine-NS particles characterized by trimethylamine ion fragments ($58[C_3H_8N]^+$ and $59[C_3H_9N]^+$) (Bhave et al., 2002; Sodeman et al., 2005; Angelino et al., 2001) showed rapid increases in number concentrations as the increase of RH (Zhong et al., 2022). Therefore, K-Amine-NS was most likely from the aging process of primarily emitted particles (e.g., traffic emissions) or mixing with secondary components during snowfall periods (Zhong et al., 2022; Cheng et al., 2018), while it was mainly relevant to regional transport during other periods (Angelino et al., 2001; Chen et al., 2020; Chen et al., 2019). As typical secondary particles, K-Amine-NS (135.9 % increase, Fig. 7) and KAECOC-NS (38.8 %) showed pronounced increases as the increase of $PA_{sulfate}/PA_{nitrate}$ during OWG.

Over 70 % of rich-Fe particles were captured during polluted periods with high humidity (~83 % on average) and occasional snowfall. Aerosol acidification associated with urban pollutants has been well documented to play a substantial role in increasing the solubility of Fe-containing particles (Rubasinghege et al., 2010; Baker and Croot, 2010; Hand et al., 2004; Zhang et al., 2014). Thus, rich-Fe particles were related to the mixing of Fe-containing particles from anthropogenic emissions (e.g. vehicle, coal combustion) with acidic salts under high relative humidity conditions (Zhu et al., 2022).

The diurnal variation of KNa-N particles showed a clear bimodal character with high values peaking at 12:00 and 21:00 (Fig. 5n), which was similar to that of COA. This result indicates that KNa-N particles are mainly from cooking fume exhaust and photochemical processes (See and Balasubramanian, 2008; Abdullahi et al., 2013; Ito et al., 2016). Similarly, KOC-NS exhibited distinct cooking characteristics and good correlation with COA ($r = 0.72$), suggesting the sources of diverse meat cooking emissions, especially the Chinese style cooking (He et al., 2004; Zhao et al., 2007).

## 3.3 Effective density of aerosol particles

The average $\rho_{eff}$ of aerosol particles for all measured sizes was 1.20 g cm$^{-3}$ with a higher value during OWG than nOWG (1.26 vs. 1.15 g cm$^{-3}$, Fig. S5a), and the $\rho_{eff}$ varied largely for different types of particles. The highly aged KAECOC-NS particles showed the highest $\rho_{eff}$ of 1.62 g cm$^{-3}$ (Fig. 10j) with abrupt increase in particle counts during snowfall and high RH period (> 70 %, Fig. 8b), while the average $\rho_{eff}$ of pure-EC and HOM associated with fresh primary emissions were 0.36 and 0.87 g cm$^{-3}$, respectively. Where the low $\rho_{eff}$ of HOM particles associated with traffic and coal combustion and containing fragments of PAHs is also attributed to their loose structure. As shown in Fig. S5, the $\rho_{eff}$ was characterized by a clear Gaussian distribution with the peak located at 1.3 g cm$^{-3}$. The $\rho_{eff}$ was proportional to ambient particle size ($r^2 = 0.93$, Fig. S5b), which was distinct from that of fresh vehicle emissions studied by Su et al. (2021). In addition, it was found by comparing the $\rho_{eff}$ of particles in different periods that the majority of particles have higher $\rho_{eff}$ during the OWG period, with the most pronounced changes in EC-NS (OWG vs. nOWG: 1.10 vs. 0.99 g cm$^{-3}$) and ECOC-NS (1.22 vs. 1.15 g cm$^{-3}$) (Fig. S6). The

average $\rho_{eff}$ of fresh and fractal-structured pure-EC particles was the same (0.36 g cm$^{-3}$) at different periods. The $\rho_{eff}$ of the primary emitted KECOC-NS (1.31 vs. 1.30 g cm$^{-3}$) and KOC-N (1.03 vs. 1.00 g cm$^{-3}$), which are closely related to the FFBBOA (Table S4) and have similar daily trends, do not change significantly between the two periods either. Irregular primary particles with low densities mix with other compositions into more compact spherical structures after being emitted into the atmosphere (Liu et al., 2019; Hu et al., 2021a).

We further analyzed the distribution of Total-EC and Total-ECOC particles as a function of $\rho_{eff}$ which contributed 56.4 % and 59.1 % at low and high $\rho_{eff}$ respectively (Fig. 9a). All EC classes except pure-EC showed bimodal distribution characteristics (Fig. 9c), including porous aggregates with low $\rho_{eff}$ peaking at 0.8 g cm$^{-3}$ and dense particles with higher $\rho_{eff}$ peaking at 1.4 g cm$^{-3}$. The results are consistent with the conclusions of previous studies (Rissler et al., 2014; Liu et al., 2019; Ma et al., 2020; Hu et al., 2022). The low $\rho_{eff}$ of pure-EC (0.36 g cm$^{-3}$ on average) suggested the presence of fresh irregularly shaped EC particles from fuel combustion. For instance, the $\rho_{eff}$ was comparable to that of diesel exhaust particles (0.25 g cm$^{-3}$, Qiu et al. (2014), fresh soot particles (0.39 g cm$^{-3}$, Rissler et al. (2014)), and propane flame particles (0.18 g cm$^{-3}$, Xue et al. (2009)). We also observed a considerable fraction of KNaECOC-NS (11.6 %) at low $\rho_{eff}$ (< 0.2 g cm$^{-3}$), indicating the major sources of solid fuel combustion. Comparably, ECOC-NS and KECOC-NS dominated the total carbonaceous particles (61.7 %) during periods with high $\rho_{eff}$ (> 1 g cm$^{-3}$).

The temporal variations of $\rho_{eff}$ are substantial due to the changes in chemical composition and particle mixing state, but overall in the range of 0.7−1.7 g cm$^{-3}$ for 77 % particles. As shown in Fig. 10a-e, the $\rho_{eff}$ of most particles was relatively stable throughout the day with slightly higher values at ~16:00. Pure-EC was significantly different from the other particles with the $\rho_{eff}$ of only 0.27 g cm$^{-3}$ at 10:00. This was attributed to the large amount of fresh elemental carbon particles during the morning rush hours. HOM, another class of particles related to fossil fuel combustion showed overall low $\rho_{eff}$ (~0.90 g cm$^{-3}$), yet with a clear increase during daytime.

The PM dependence of $\rho_{eff}$ is shown in Fig. 10. It is clear that the $\rho_{eff}$ of almost all types of aerosol particles increased as a function of PM levels. For example, the average $\rho_{eff}$ of Total-EC increased from 1.01 g cm$^{-3}$ during clean periods (NR-PM$_1$ < 10 μg m$^{-3}$) to 1.28 g cm$^{-3}$ during polluted periods (NR-PM$_1$ > 50 μg m$^{-3}$), indicating the formation process of pollution also led to the changes in both aerosol composition and particle density. Carbonaceous particles contributed up to 79.8 % of the total particles during the polluted period (Fig. 8e), with the most abundant EC-NS and KECOC-NS particles having an average density of 1.18 and 1.70 g cm$^{-3}$ (NR-PM$_1$ > 50 μg m$^{-3}$), respectively. The changes of $\rho_{eff}$ as a function of RH were similar considering that severe pollution in winter was generally associated with high RH. We noticed that the average $\rho_{eff}$ was minimal during periods with RH = 40−50 %, which was 1.09 g cm$^{-3}$, coincident with the period with NR-PM$_1$ in the range of 20–30 μg m$^{-3}$. The proportion of primary OA in total OA in this case was considerable (47.7 % and 40.0 %, Fig. 8d, h), indicating that the appearance of low $\rho_{eff}$ was mainly caused by fresh particles at moderate RH and PM concentration. The high $\rho_{eff}$ (1.36 g cm$^{-3}$) corresponding to RH above 70 % was mainly associated with the pollution during the snowfall and the formation of secondary OA, which accounted for 85.5 % of OA (Fig. 8d). Particularly, the $\rho_{eff}$ of K-Amine-NS increased by more than 30 % as the increase of RH, highlighting the formation of N-containing particles through aqueous-phase processing and the change of particle density as well (Zhong et al., 2022).

## 4 Conclusions

This study provides a detailed analysis of chemical composition, mixing state, and effective density of ambient aerosol particles during Olympic Winter Games in Beijing by using a DMA/AAC+HR-SPAMS tandem system. Nearly 760,000 particles were classified into seven major classes including Total-EC (13.4 %), Total-ECOC (47.1 %), Total-OC (10.5 %), Total-IA (7.0 %), Biomass-K (13.0 %), HOM (5.5 %), and Metals (3.5 %). 71.0 % of particles were found to be carbonaceous particles mixed primarily with sulfate and nitrate. The emission controls during OWG led to decreases in the types of Total-EC particles from 11.5 % to 8.5 %, yet increases in aged and regional particles, e.g., ECOC-NS. The average effective density of aerosol particles between 150 and 300 nm was 1.20 g cm$^{-3}$, with higher values during OWG (1.26 g cm$^{-3}$ vs. 1.15 g cm$^{-3}$ for nOWG). The Total-EC particles dominated the total particles at low effective densities (56.4 %), and the effective densities of EC class particles except pure-EC showed a bimodal distribution peaking at 0.8 and 1.4 g cm$^{-3}$, respectively. Comparatively, the fresh pure-EC showed much lower density with an average effective density of 0.36 g cm$^{-3}$. The effective density varies largely depending on the particle types, secondary formation, and the changes in RH. Overall, high effective densities usually occur during the periods with high PM (NR-PM$_1$ > 50 μg m$^{-3}$) and RH (> 70 %), highlighting the impact of aging processes on the effective density of aerosol particles. It is worth mentioning that due to the different sensitivities of the ionizing laser of SPAMS for the detection of different chemical compositions, its capability of quantitative analysis needs to be further evaluated in future studies.

**Data availability.** The data in this study are available from the authors upon request (sunyele@mail.iap.ac.cn).

**Author contributions.** YS and AD designed the research. AD, HL, WX, YZ, JS, YL and WZ conducted the measurements and experiments. AD, WX, HL, YZ and WZ analyzed the data. LL, XD, XP and ZW reviewed and commented on the paper. AD, YS and JS wrote the paper.

**Competing interests.** The contact author has declared that none of the authors has any competing interests.

**Acknowledgements.** This work was supported by the National Natural Science Foundation of China (92044301).

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

**Table 1: A summary of particle types during this campaign.**

| | Classification of particles | Number | Percentage (%) |
|---|---|---|---|
| Total-EC | pure-EC | 3317 | 0.45 |
| | EC-NS | 48606 | 6.57 |
| | KEC-N | 26250 | 3.33 |
| | KNaEC-N | 21076 | 2.85 |
| Total-ECOC | ECOC-NS | 108490 | 14.67 |
| | KECOC-CN | 8493 | 1.15 |
| | KECOC-NS | 170590 | 23.06 |
| | KNaECOC-NS | 35821 | 4.84 |
| | KAECOC-NS | 23976 | 3.24 |
| Total-OC | KOC-N | 38376 | 5.19 |
| | KOC-NS | 31240 | 4.22 |
| | K-Amine-NS | 7665 | 1.04 |
| Total-IA | K-N | 35946 | 4.85 |
| | KNa-N | 15908 | 2.15 |
| Biomass-K | | 96153 | 13.00 |
| High-molecular-weight organic matter (HOM) | | 40998 | 5.54 |
| Metals | rich-Fe | 15562 | 2.10 |
| | other | 10787 | 1.44 |

**Table 2: Comparison of the average ($\pm\sigma$) meteorological parameters and pollutant concentrations during OWG and nOWG period.**

| | $T$ (°C) | RH (%) | WS (m s$^{-1}$) | NR-PM$_1$ (µg m$^{-3}$) | eBC (µg m$^{-3}$) |
|---|---|---|---|---|---|
| **OWG** | -1.2±4.4 | 32.0±18.1 | 3.6±1.6 | 7.8±7.4 | 1.0±0.8 |
| **nOWG** | 0.9±4.4 | 36.0±23.7 | 3.4±2.0 | 15.2±19.5 | 1.6±1.7 |

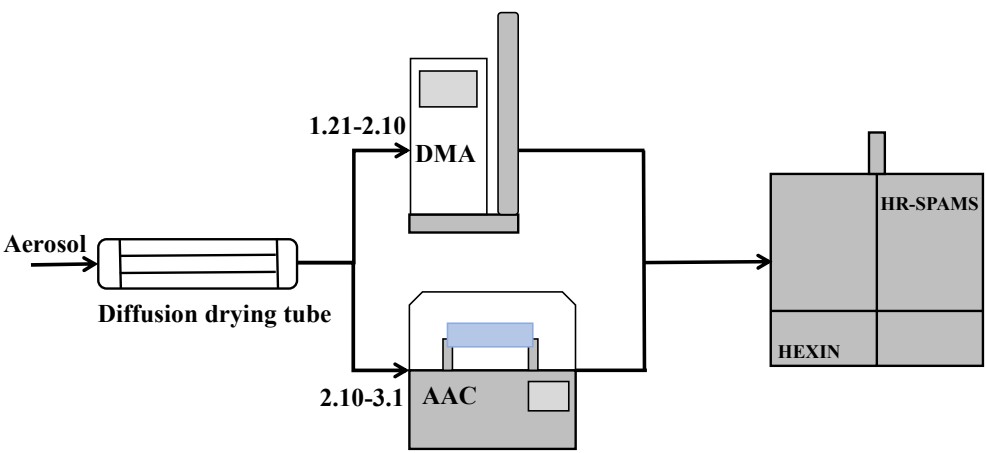

**Figure 1: Schematic diagram of the experimental system.**

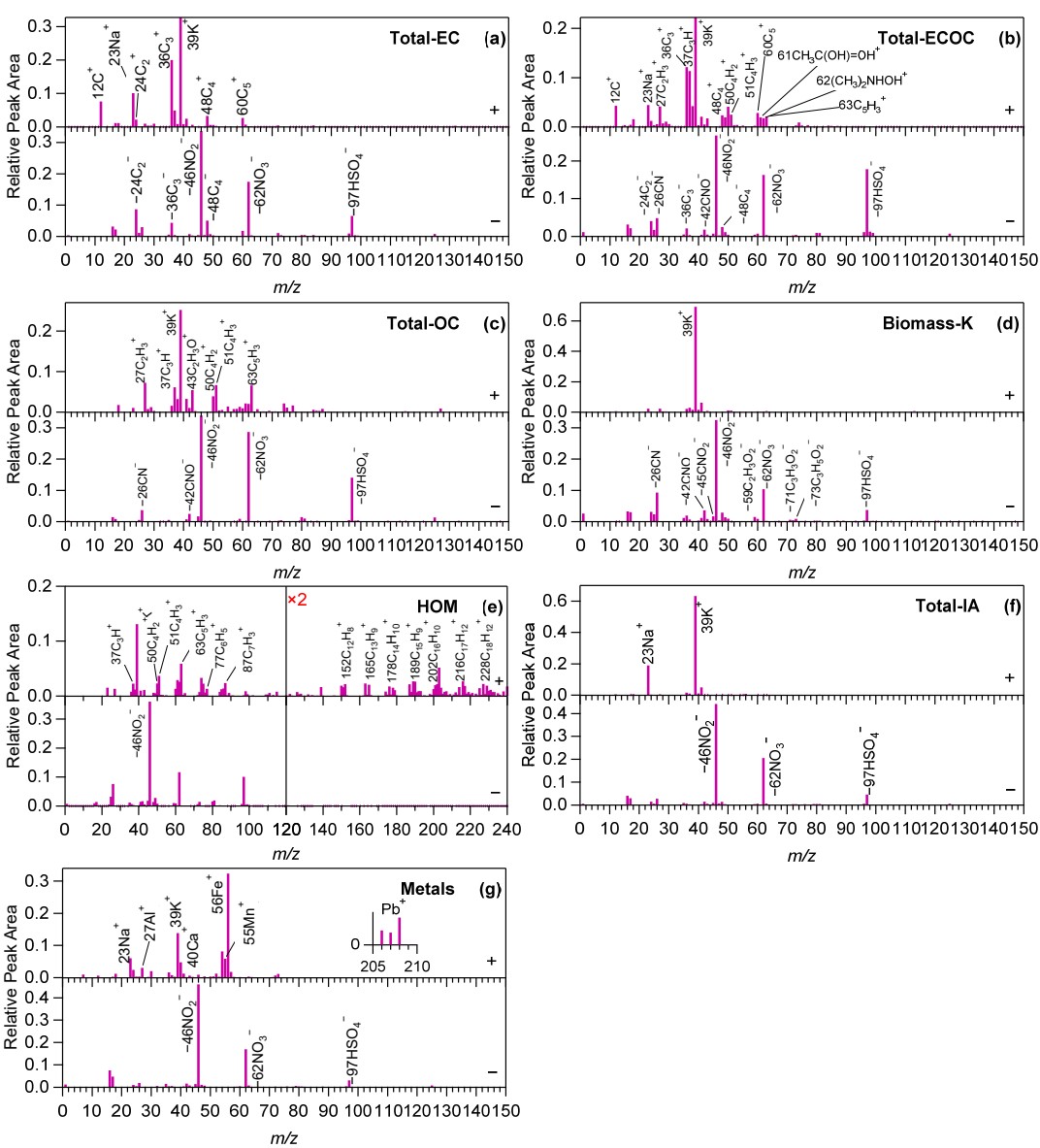

**Figure 2: Average mass spectra of single particles for seven major classes.**

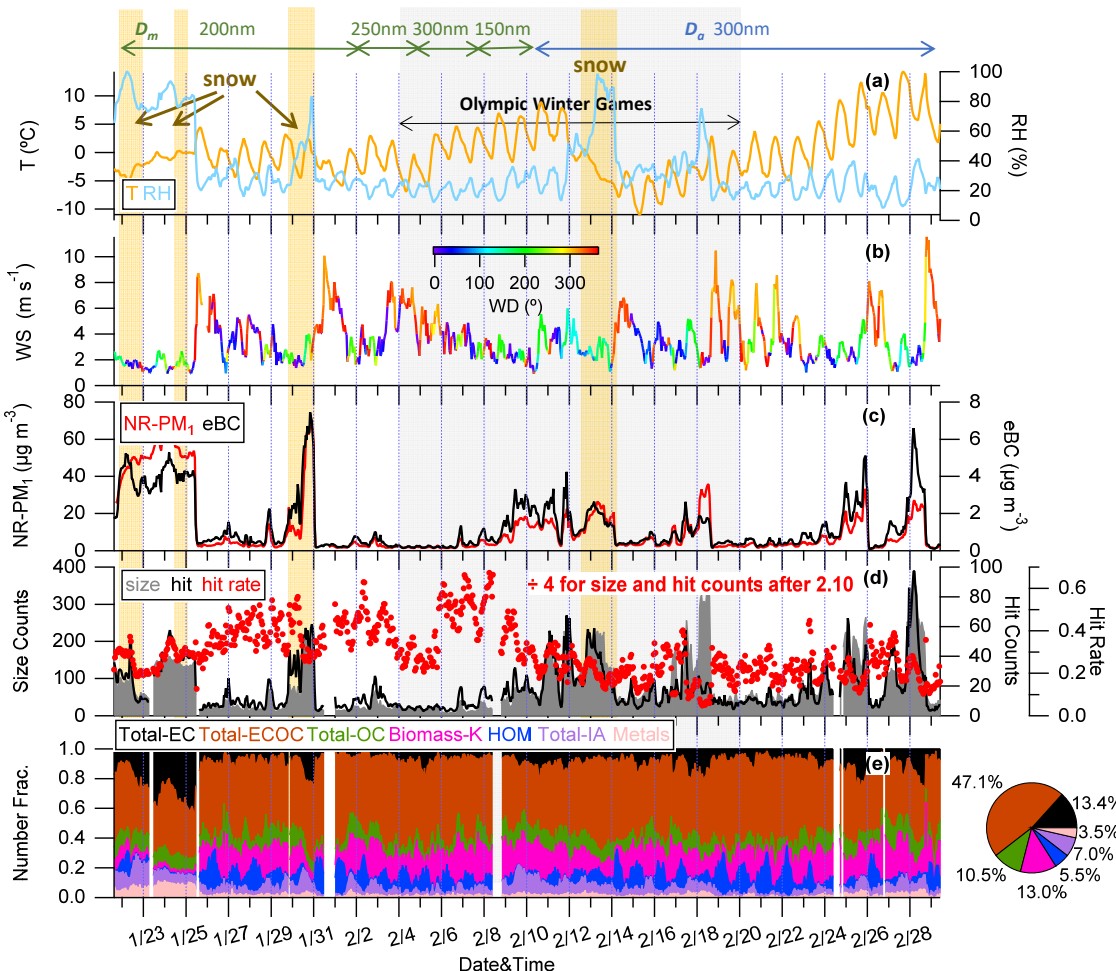

**Figure 3: Time series of (a) temperature (*T*) and relative humidity (RH), (b) wind speed (WS) colored by wind direction (WD, measured in degrees clockwise from due north), (c) mass concentrations of eBC and NR-PM₁, (d) number of sized particles, hit particles as well as the average hit rate of SPAMS per minute (both size and hit counts after 2.10 are divided by 4 in order to show the temporal trend of the DMA-SPAMS period better) and (e) number fraction of seven major classes of particles including Total-**

545 **EC, Total-ECOC, Total-OC, Total-IA, Biomass-K, HOM and Metals. The pie chart shows the average number fraction of particles for the entire period. The green and blue arrows represent the sizes selected by DMA or AAC, respectively, while the yellow and gray shading corresponds to the snowfall and Olympic Winter Games periods, respectively.**

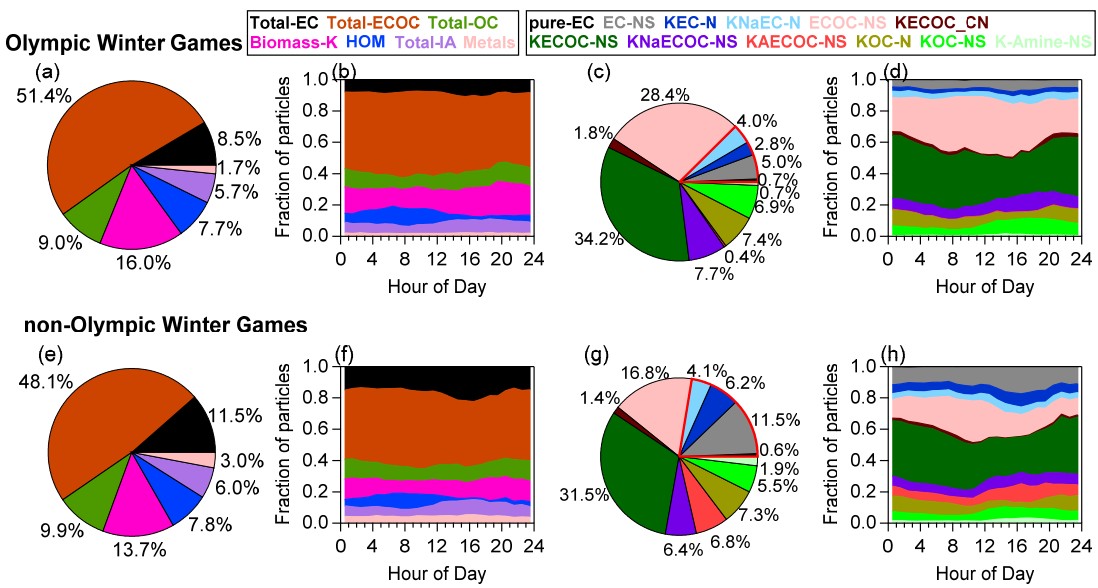

Figure 4: Average contributions and diurnal fractional contributions of the major types and carbonaceous particles during (a-d) Olympic Winter Games and (e-h) non-Olympic Winter Games.

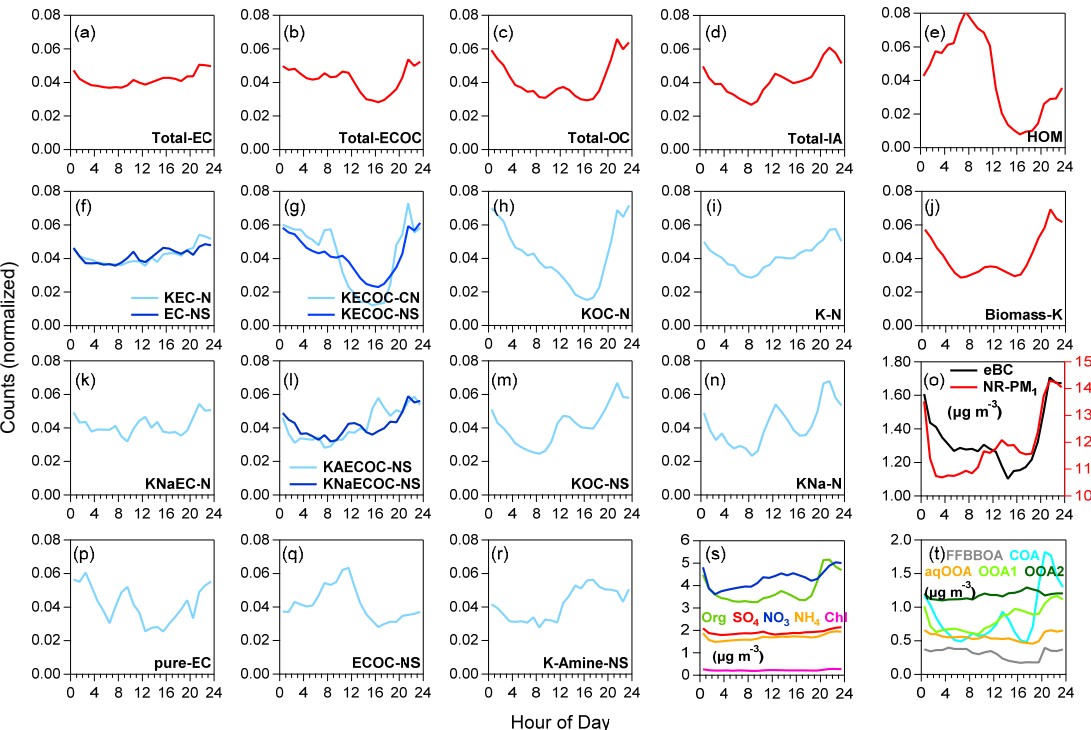

**Figure 5: Diurnal cycles of (a-n, p-r) normalized counts for each class of particles, (o) mass concentrations of eBC and NR-PM₁, and (s) NR-PM₁ species, and (t) OA factors.**

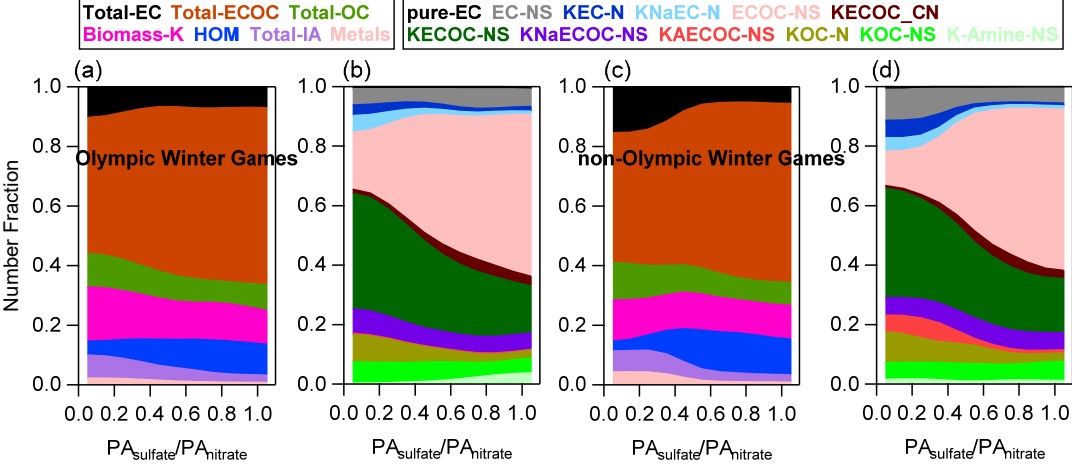

**Figure 6: Variations of the average contributions of major types and carbonaceous particles as a function of peak area ratio of sulfate to nitrate during (a, b) OWG and (c, d) nOWG.**

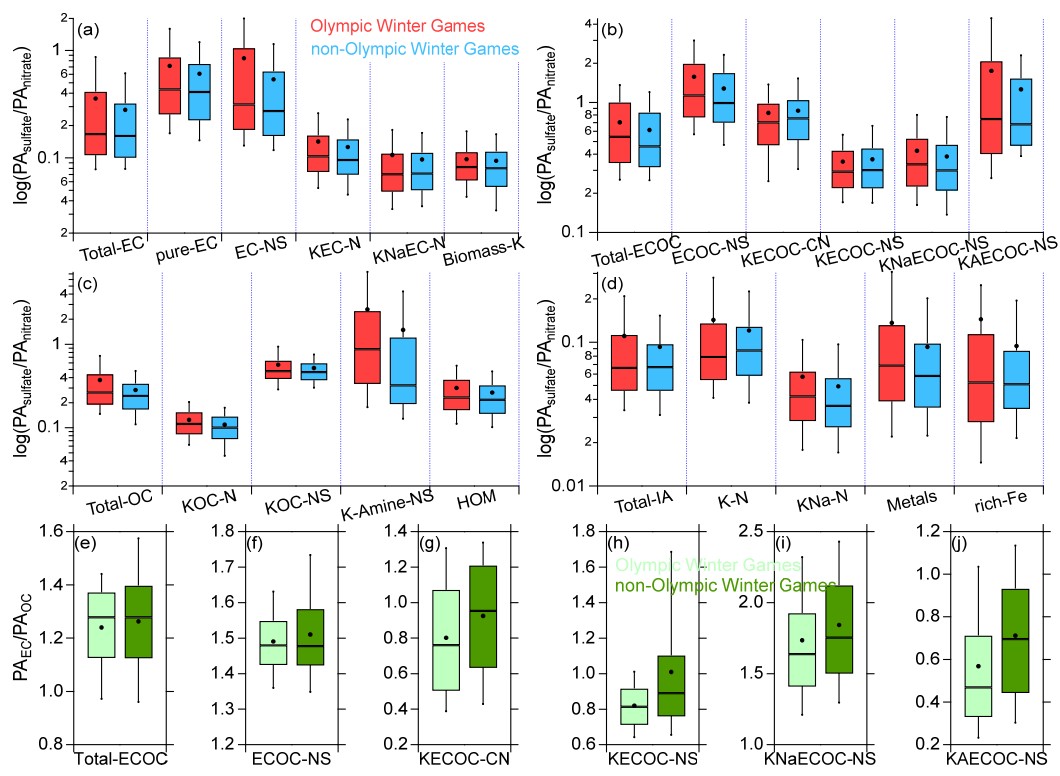

**Figure 7: Peak area ratios of (a-d) sulfate (m/z −80 and −97) to nitrate (m/z −46 and −62) for each type of particles and (e-j) elemental carbon (m/z $C_n^{\pm}$, n = 1−5) to organic carbon (m/z 27, 29, 37 and 43) in ECOC-containing particles during OWG and nOWG. Also shown are median (horizontal lines), mean (circles), 25th and 75th percentiles (lower and upper boxes), and 10th and 90th percentiles (lower and upper whiskers).**

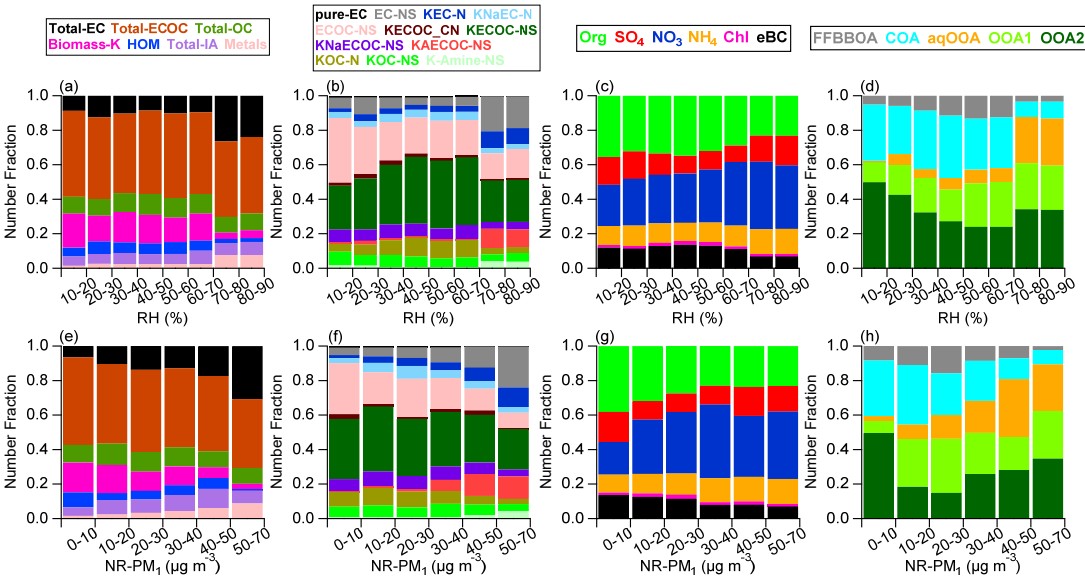

**Figure 8: Variations of the number fractions and mass fractions of aerosol particles and species as a function of (a-d) relative humidity and (e-h) NR-PM$_1$ mass concentrations.**

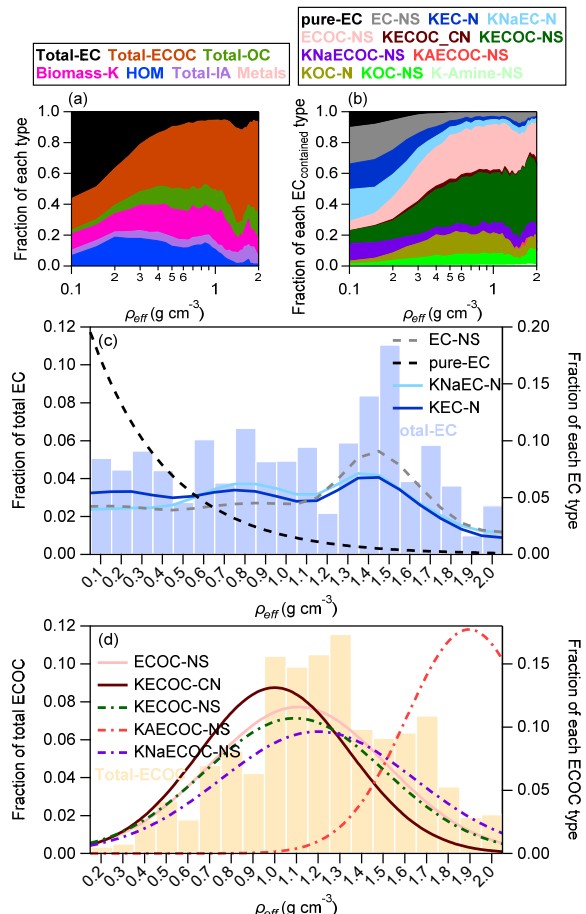

570

**Figure 9:** Average contributions of (a) major types and (b) carbonaceous particles as a function of $\rho_{eff}$. The distributions of $\rho_{eff}$ for (c) Total-EC and (d) Total-ECOC particles. The left y–axis is applied to the column diagram and the right y–axis is applied to the density fitting curves.

575

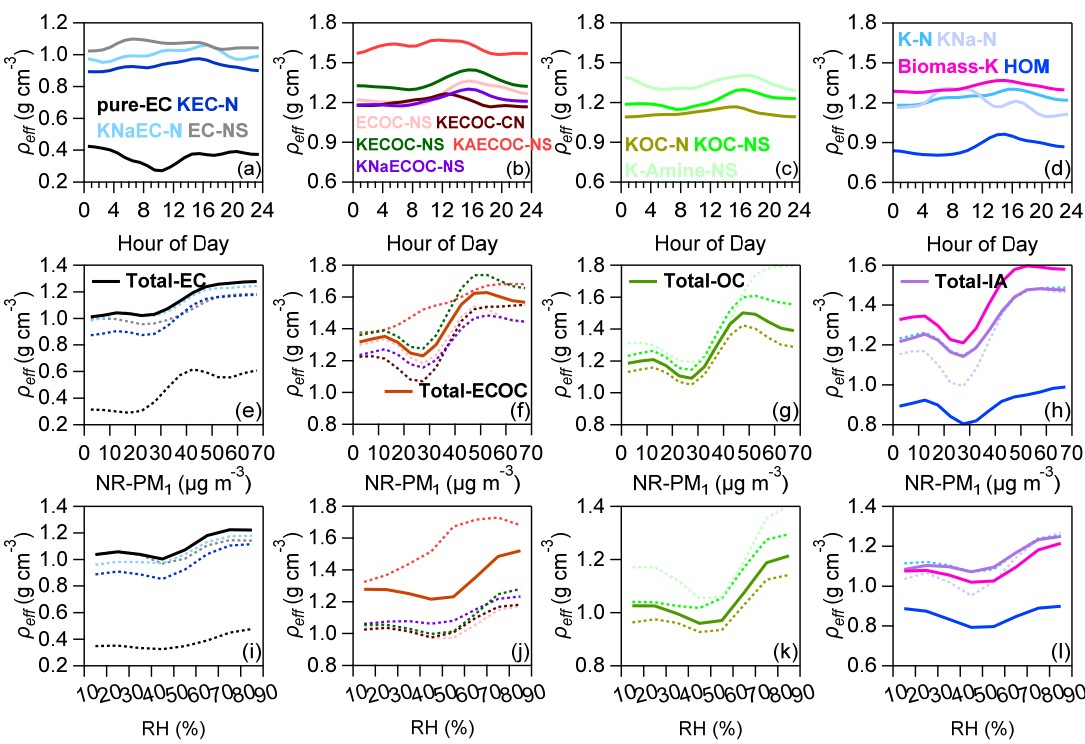

**Figure 10: Diurnal variations of $\rho_{eff}$ of each particle species (a-d). And variations of $\rho_{eff}$ as a function of (e-h mass concentration of NR-PM$_1$ and (i-l) relative humidity. Where the $\rho_{eff}$ for each particle type is averaged over the sum of all $D_m$ and $D_a$ sizes.**