# Peer review of "Mixing state and effective density of aerosol particles during the Beijing 2022 Olympic Winter Games"

_EGUsphere, 2023_

## Author Comment (AC1)

We are thankful to the two reviewers for their thoughtful and constructive comments that help improve the manuscript substantially. We have revised the manuscript accordingly. Listed below is our point-to-point response in blue to each comment that was offered by the reviewer.

**Response to Reviewer #1**

This study used different types of instruments with mass spectra, combining with a DMA and AAC for size selection and investigated the effective density of aerosols, during the Beijing 2022 Olympic Winter Games for the impacts of emission controls on particle mixing state and density. The results provide information on changes in aerosol compositions and mixing state due to emission control. A few points need to be addressed before it can be accepted.

We thank the reviewer's comments and have revised the manuscript accordingly.

**Major:**

1) The method in deriving the effective density for different compositions should be given more details. The used equations have not been clearly explained. It is not clear how you have linked the effective density with certain composition.

Thank the reviewer's comments. SPAMS can provide information on the mixing state and particle size, i.e., the vacuum aerodynamic diameter ($D_{va}$), of individual particles, while DMA and AAC can select particles with specific mobility diameters ($D_m$) or aerodynamic diameters ($D_a$), respectively. The relationship between the three diameters has been given by Decarlo et al. (2004) and the effective density ($\rho_{eff}$) can be calculated if either two of them are known. For example, when the $D_{va}$ and $D_m$ of the particle are known, the $\rho_{eff}$ can be calculated as:

$$\rho_{eff} = \frac{D_{va}}{D_m} \rho_0 \tag{1}$$

It can be used to calculate the $\rho_{eff}$ of particles captured by the DMA-SPAMS tandem system. Where $\rho_0$ is the standard density (1.0 g cm$^{-3}$). Another approach to define the $\rho_{eff}$ that can be adopted in the AAC-SPAMS tandem system is based on the ratio of particle density ($\rho_p$) and the particle dynamic shape factor ($\chi_\gamma$) as follows:

$$\rho_{eff} = \frac{\rho_p}{\chi_\gamma} = \frac{D_{va}}{D_{ve}\rho_0} \tag{2}$$

where $D_{ve}$ represents the volume equivalent diameter. The method of deriving the $\rho_{eff}$ of particles with the support of $D_{ve}$ and $D_{va}$ has been verified in detail in previous studies (Peng et al., 2021; Su et al., 2021). The relationship between the $D_a$, $D_{va}$ and $D_{ve}$ can be stated by the following equation:

$$D_a = D_{ve}\sqrt{\frac{\rho_p C_c(D_{ve})}{\chi_t \rho_0 C_c(D_a)}} \tag{3}$$

where $\chi_t$ represents the aerosol dynamic shape factor in the transition regime. Considering the

approximation between $\chi_t$ and $\chi_\gamma$, the $D_{ve}$ can be calculated by combining Eqs. (2) and (3) as follows:

$$C_c(D_a)\frac{D_a^2}{D_{va}} = D_{ve}C_C(D_{ve}) \qquad (4)$$

That is, when the aerosol instruments in tandem are same (DMA-SPAMS or AAC-SPAMS), the derivation of $\rho_{eff}$ of particles with different compositions is uniform and its confidence has been confirmed in previous studies (Su et al., 2021; Spencer et al., 2007; Peng et al., 2021). Based on the diameter values set by DMA or AAC, combined with the $D_{va}$ and chemical compositions of the particles provided by SPAMS, it is possible to associate the $\rho_{eff}$ of individual particles with their chemical compositions. Moreover, we have made additional explanations in lines 123–124 of the manuscript and given detailed calculation methods of $\rho_{eff}$ in the supplementary according to the recommendations.

2) The instrument setup should be given in front in the main texts, with more explanation why running AAC and DMA in parallel. Why the density has been derived using two methods.

Thank the reviewer's comments. In fact, we initially planned to connect DMA and AAC in series with SPAMS at different periods and select particles with $D_m$ and $D_a$ in the range of 150–300 nm and 200–700 nm, respectively, to finally obtain two complete datasets. However, only the SPAMS data with $D_a = 300$ nm were eventually credible in the AAC-SPAMS period, accounting for 13.3% of the total particles captured by SPAMS (322415 of 2416964). This is due to the unstable sheath flow of AAC when selecting particles in the size range of 400–700 nm, and only 1756 particles were captured at $D_a = 200$ nm due to the SPAMS detection limit. We therefore decided to use the DMA-SPAMS in combination with the AAC-SPAMS dataset, which covers the Olympic Winter Games completely and makes it possible to analyze the changes in the mixing state and $\rho_{eff}$ of particles under emission control. Considering that DMA and AAC screen particles based on different diameters, it is necessary to calculate the $\rho_{eff}$ by two methods separately. We have added the experimental system schematic (Fig. 1) to the manuscript as suggested and made additional explanations about the calculation of $\rho_{eff}$ in lines 114-116.

3) The effective density from 1.26 to 1.20 is not significantly different, as emphasized in the abstract.

Thank the reviewer for pointing this out. The 1.20 and 1.26 g cm$^{-3}$ in the abstract correspond to the average $\rho_{eff}$ of particles for the entire period and for the OWG, respectively, which are indeed similar. In order to emphasize the change in particle aging due to reduced emissions, it may be more convincing to compare the $\rho_{eff}$ of particles during the OWG with the nOWG (1.26 vs. 1.15 g cm$^{-3}$). Actually, the range of average $\rho_{eff}$ for all particles in this study is wide (0.76–1.68 g cm$^{-3}$) and is influenced by the chemical composition and atmospheric processes. Meanwhile, the $\rho_{eff}$ ranges for different classes of particles (from 0.36 g cm$^{-3}$ for pure-EC to 1.62 g cm$^{-3}$ for KAECOC-NS) are also comparable to the results of previous studies in Guangzhou (0.87–1.51 g cm$^{-3}$) and California (0.27–1.48 g cm$^{-3}$) (Spencer et al., 2007; Zhang et al., 2016). The relatively small difference in $\rho_{eff}$ between the OWG and nOWG may be attributed to the fact that primary emitted KECOC-NS (1.31 vs. 1.30 g cm$^{-3}$) and Biomass-K (1.10 vs. 1.08 g cm$^{-3}$) particles, which contributed up to 36.06% of the total particles, had very little change in $\rho_{eff}$ between the two periods (Fig. S6). In addition, the

large number of KAECOC-NS particles with high $\rho_{eff}$ captured during snowfalls on 22 January, 24 January, and 30 January also led to small difference in $\rho_{eff}$ between the OWG and nOWG periods (for KAECOC-NS, OWG vs nOWG: 1.36 vs 1.71 g cm$^{-3}$). We have added Fig. S6 to the supplementary in order to emphasize the difference in $\rho_{eff}$ between the OWG and nOWG periods. Moreover, additional explanation on $\rho_{eff}$ has been added in lines 234-241, while lines 23-24 of the abstract have been revised.

[Figure]

Figure S6: Average effective density of different classes of particles during the OWG and nOWG periods.

4) Why high molecular weight OA has a lower effective density.

As shown in Fig. R1a, the average mass spectra of high-molecular-weight organic matter (HOM) exhibits distinct fragments of polycyclic aromatic hydrocarbons (PAH) such as 152 $[C_{12}H_8]^+$, 165 $[C_{13}H_9]^+$, 178 $[C_{14}H_{10}]^+$, and 189 $[C_{15}H_9]^+$. Previous studies based on single particles have demonstrated that particles with significant fragments of PAHs in urban areas are mostly associated with vehicle emissions and coal combustion (Sodeman et al., 2005; Su et al., 2021; Giorio et al., 2015; Hu et al., 2021; Wang et al., 2020; Zhang et al., 2022). The lower PA$_{sulfate}$/PA$_{nitrate}$ ratio (0.24) indicates that HOM belongs to relatively fresh particles that have not been sufficiently aged. Meanwhile, the peak of the daily cycle of HOM at 8:00 and the high values at night (Fig. R1b) coincide with the traffic emission during the morning rush hours and coal combustion at night, supporting the conclusion that HOM particles are comparatively fresh.

Effective density ($\rho_{eff}$), as a parameter often derived from a combination of two aerosol measurements, can be defined as the ratio of the particle density ($\rho_p$) to the bulk material density ($\rho_m$) (Hand et al., 2002; Mcmurry et al., 2002). $\rho_p$ is less than $\rho_m$ when the particles contain voids inside, with the $\rho_{eff}$ of particles less than 1 (Decarlo et al., 2004). It is recognized that fresh particles emitted by combustion usually show loose structure with irregular shape (Liu et al., 2019; Spencer et al., 2007; China et al., 2014). That is, HOM particles emitted from fossil fuel combustion will have lower $\rho_{eff}$ due to their loose structure, which is consistent with our results (0.87 g cm$^{-3}$ on average). In addition to particle morphological characteristics, the $\rho_{eff}$ of HOM is also affected by factors including chemical composition and aging process (Katrib et al., 2005; Pagels et al., 2009). As HOM emitted into the atmosphere undergoes the aging process and mixes with more sulfate and nitrate, they become more compact and the $\rho_{eff}$ peaks at 14:00 (0.96 g cm$^{-3}$). Thanks to the reviewers

for pointing this out, and we have provided additional explanation in lines 231-232.

[Figure]

Figure R1: Average mass spectra of single particles (a) and diurnal cycle of normalized counts (b) for HOM.

5) What can the findings tell from environmental policy point of view?

Thank the reviewer's comments. This study investigated the effects of emission control measures on atmospheric particles during major events by comparing the mass concentration, chemical composition, and effective density of particles during the Olympic Winter Games with other periods. The results indicate that significant improvements in air quality can be achieved in the short term by closing high-emission plants and limiting high-emission vehicles. Control measures during the OWG resulted in a 48.7% and 37.5% decrease in mass concentrations of NR-PM$_1$ and eBC particles, respectively. It is worth noting that the short-term approaches to improving air quality also provide new insights into the development of environmental policy. Due to the implementation of desulphurization, accompanied by increasing NOx emissions from vehicles and industry, nitrates became the most dominant pollutant in urban areas, accounting for 36.1% of NR-PM$_1$ during this campaign. Therefore, environmental policies need to be formulated not only in terms of its long-term viability, but also with a focus on the emission and control of NO$_x$. Utilizing clean energy instead of traditional fossil fuels as much as possible in the production of factories, and further raising the emission standards for factory waste and encouraging the purchase and use of waste treatment equipment. In addition, highly polluting and emitting fuel vehicles can be eliminated gradually while new energy vehicles are actively promoted.

**Others:**

Please indicate the full name where the abbreviation first appears in the abstract, such as EC and OC on line 17, EC-NC and KEC-N in line 19, and ECOC-NC in line 20.

Thank the reviewer's comments. We have revised the manuscript in lines 17-21 as suggested.

Please give the full name of NR-PM1 when it appears for the first time.

Changed in lines 101-102 as suggested.

Line105-114, Why to use DMA connecting with SPAMS and AAC to connect with SPAMS separately to obtain effective particle size? Why are two different instruments required?

Thank the reviewer's comments. As mentioned in main question 2, we had originally planned to connect DMA and AAC in series with SPAMS at different periods and select particles with $D_m$ and $D_a$ in the range of 150–300 nm and 200–700 nm, respectively, to finally obtain two complete datasets. Unfortunately, only the SPAMS data with Da = 300 nm were eventually credible in the AAC-SPAMS period, accounting for 13.3% of the total particles captured by SPAMS (322415 of 2416964). Therefore, in order to make the conclusions of this study more convincing, we decided to use the DMA-SPAMS in combination with the AAC-SPAMS dataset, which fully covers the Olympic Winter Games and makes it possible to analyze the changes in the mixing state and effective density of particles under emission control. Considering that the $\rho_{eff}$ is usually defined by the combination of two aerosol size measurements (Hand et al., 2002; Mcmurry et al., 2002), two formulas, i.e., Eqs. (1) and (2) above, need to be used in the calculation of $\rho_{eff}$.

Could you provide a detailed explanation how Equation 2 is derived?

Thanks to the reviewer's suggestion, we have provided more detailed derivations both below and in Section 1.2 of the supplementary.

According to the definition given by Hand and Kreidenweis (2002), the effective density of particles is equal to the ratio of the particle density ($\rho_p$) to the dynamic shape factor ($\chi_\gamma$).

$$\rho_{eff} = \frac{\rho_p}{\chi_\gamma} \tag{5}$$

The calculation of $D_{va}$ obtained by Jimenez (2003) is shown in Eq. (6).

$$D_{va} = \frac{\rho_p D_{ve}}{\rho_0 \chi_\gamma} \tag{6}$$

Combining Eqs. (5) and (6) to obtain the following formula for $\rho_{eff}$, which is mentioned by the reviewer:

$$\rho_{eff} = \frac{D_{va}}{D_{ve}\rho_0} \tag{7}$$

Besides, the relationship between the $D_a$, $D_{va}$ and $D_{ve}$ can be stated by Eq. (8):

$$D_a = D_{ve}\sqrt{\frac{\rho_p C_c(D_{ve})}{\chi_t \rho_0 C_c(D_a)}} \tag{8}$$

where $\chi_t$ represents the aerosol dynamic shape factor in the transition regime. Considering the approximation between $\chi_t$ and $\chi_\gamma$, Eqs. (6) and (8) can be combined and calculated as follows:

$$C_c(D_a) \frac{D_a^2}{D_{va}} = D_{ve} C_C(D_{ve}) \tag{9}$$

$C_c(D)$ is the Cunningham slip correction factor, which can be calculated by the following equation:

$$Cc(D) = 1 + \frac{\lambda}{D}(A + B \cdot exp(\frac{C \cdot D}{\lambda})) \tag{10}$$

where $\lambda$ represents the mean free path of the gas molecules. A, B and C are empirically determined constants specific to the analyzed system, where A is 2.33, B is 0.966 and C is -0.498. Substituting Eq. (10) into Eq. (9) obtains Eq. (11):

$$\frac{D_a^2}{D_{va}} + \frac{D_a \cdot \lambda}{D_{va}}\left(A + B \cdot exp\left(\frac{C \cdot D_a}{\lambda}\right)\right) = D_{ve} + \lambda\left(A + B \cdot exp\left(\frac{C \cdot D_{ve}}{\lambda}\right)\right) \tag{11}$$

The $D_a$ and $D_{va}$ are known in the AAC-SPAMS tandem system, which can be brought into Eq. (11) to obtain $D_{ve}$. Finally, the $\rho_{eff}$ of particles can be derived from Eq. (7).

Line 212-214, Please indicate the corresponding figure number for the conclusive numerical results provided by the authors.

Added as suggested.

Please specify what the color bar in Figure S5 stands for.

Thanks to the reviewer for pointing this out, an explanation has been added in the figure legend of Fig. S5.

Section 3.3 is quite confused. The title indicates that the study focused on the effective density of aerosols during the Olympic Winter Games. However, the effective density is only briefly mentioned but not related to the event.

Thanks to the reviewer for pointing this out. We further compare the average $\rho_{eff}$ of different classes of particles during the OWG and nOWG periods as shown in Fig. S6. The results show that most of the particles have higher $\rho_{eff}$ during the OWG, with the most pronounced changes in EC-NS (1.10 vs. 0.99 g cm$^{-3}$) and ECOC-NS (1.22 vs. 1.15 g cm$^{-3}$). In contrast, the $\rho_{eff}$ of fresh pure-EC (0.36 vs. 0.36 g cm$^{-3}$), KECOC-NS (1.31 vs. 1.30 g cm$^{-3}$) and KOC-N (1.03 vs. 1.00 g cm$^{-3}$) particles from primary emission did not change significantly in both periods. The KAECOC-NS particles with significantly high $\rho_{eff}$ during the nOWG period were affected by snowfall, and their $\rho_{eff}$ increased from 1.32 to 1.73 g cm$^{-3}$ as RH increased from 10% to 80% (Fig. 9j). We have added Fig. S6 as suggested, with additional description in lines 234-241 of the manuscript.

As this study mainly focused on the BC-containing particles, a few quite related studies also measured the shape of BC-containing particles in Beijing, which showed more spherical particles when polluted (Hu et al., EST Letters, 2022, 10.1021/ acs.estlett.2c00060), and also the

aerodynamic size-selected compositions and density by AAC (Yu et al., ACP, 2022, 10.5194/acp-22-4375-2022). These studies could be referenced to support some of your conclusions.

Thanks to the reviewer's suggestion. The literatures were cited in the revised manuscript.

2 is not used in the texts.

Thank the reviewer's comments. The original Fig. 2 is mentioned in lines 127-131 of the manuscript in order to depict the meteorological elements, pollutant concentrations, and the particle counts captured by SPAMS throughout the observation period. The original Fig. S2 is referenced in lines 101-103 of the manuscript to illustrate the representativeness of the SPAMS measurements by comparing them to the AMS and AE33 measurements. The results of the comparison of pollutant concentrations between Winter Olympic and non-Winter Olympic periods given in Table 2 are also mentioned in lines 128-130. The characteristic peak information provided in Table S2, which is essential for particle classification, is mentioned in line 112.

Please explain the many significantly high values of PAsulfate/PAnitrate in Figure 6.

Thank the reviewer's comments. We selected $80[SO_3]^-$ and $97[HSO_4]^-$ as characteristic peaks for sulfate and $46[NO_2]^-$ and $62[NO_3]^-$ for nitrate in the calculation of $PA_{sulfate}/PA_{nitrate}$. Previous data processing results showed that the average $PA_{sulfate}/PA_{nitrate}$ values for particles including K-Amine-NS, K-N, and rich-Fe were significantly higher than the 75th percentile. Checks revealed that some of the abnormally high values of $PA_{sulfate}/PA_{nitrate}$ were not removed during previous data processing, so we further processed the data and redrew the graph (Fig. 6). However, the average $PA_{sulfate}/PA_{nitrate}$ values of some particles are still high, with K-Amine-NS and rich-Fe being the most obvious. This is due to the fact that local pollutants are removed while particles mixed with sulfate are transported during high wind speed periods, resulting in high $PA_{sulfate}/PA_{nitrate}$ values (Fig. R3). For example, the average $PA_{sulfate}/PA_{nitrate}$ for K-Amine-NS and rich-Fe for the entire period were 2.04 and 0.13, respectively, but when the wind speed was higher than 6 m s$^{-1}$ (11.4% of the entire period), they were 5.37 and 0.32, respectively (Figs. R2a and R2b). Such high $PA_{sulfate}/PA_{nitrate}$ values at high wind speeds and low pollutant concentrations lead to the average $PA_{sulfate}/PA_{nitrate}$ values in the original Fig. 6 being significantly higher than the median values.

While the differences in $PA_{sulfate}/PA_{nitrate}$ values between different classes of particles are mainly related to their mixing state. Overall, the captured particles were predominantly mixed with nitrate, with an average $PA_{sulfate}/PA_{nitrate}$ of only 0.25 during the observation period, with average $PA_{sulfate}/PA_{nitrate}$ values of 0.13, 0.10, 0.12, 0.10 and 0.15 for KEC-N, KNaEC-N, KOC-N, Biomass-K and Total-SIA, respectively.

[Figure]

Figure 6: Peak area ratios of (a-d) sulfate (m/z −80 and −97) to nitrate (m/z −46 and −62) for each type of particles and (e-j) elemental carbon (m/z $C_n^{\pm}$, n = 1−5) to organic carbon (m/z 27, 29, 37 and 43) in ECOC-containing particles during OWG and nOWG. Also shown are median (horizontal lines), mean (circles), 25th and 75th percentiles (lower and upper boxes), and 10th and 90th percentiles (lower and upper whiskers).

[Figure]

Figure R2: Time series of $PA_{sulfate}/PA_{nitrate}$ for (a) K-Amine-NS, (b) rich-Fe, (c) K-N, (d) Total-EC, (e) Total-ECOC, (f) HOM, (g) Biomass-K and (h) Total-OC, and (i) wind speed (WS) colored by wind direction (WD) as well as mass concentration of NR-PM$_1$.

[Figure]

Figure R3: Scatter plot of $PA_{sulfate}/PA_{nitrate}$ values for all captured particles versus the mass

concentration of NR-PM$_1$ and eBC measured by AMS, with scatters colored by wind speed.

**Response to Reviewer #2**

The manuscript " Mixing state and effective density of aerosol particles during the Beijing 2022 Olympic Winter Games " mainly investigates the impacts of emission controls on particle mixing state and density by deploying a single particle aerosol mass spectrometer in tandem with a differential mobility analyzer and an aerodynamic aerosol classifier during the Beijing 2022 Olympic Winter Games (OWG). In general, the paper is well written and presented in a logical way. It is of general interest for Atmospheric Chemistry and Physics related communities. I therefore recommend publication of this paper in Atmospheric Chemistry and Physics after some revisions. My comments are listed as follows:

Specific Comments:

Line 16: "showed" should be changes to "show". Besides, I suggest the authors to use present tense in writing a scientific article. I recommend to use external proof reading before submission of the revised version.

Thanks to the reviewers for pointing this out, and we have revised it as suggested.

Lines 16-17: the meaning of "Total- EC", "Total-OC" and "Total-ECOC" should be given here, also including other abbreviations (e.g., EC-NS, KEC-N, and ECOC-NS).

Thank the reviewer's comments. We have added the meaning of the abbreviations in lines 17-21.

Line: 75-76: a detailed explanation for "DMA (model 3085A, TSI Inc.) and SPAMS (Hexin Analytical Instrument Co., Ltd.), AAC (Cambustion Ltd.)" should be given.

Thank the reviewer's comments. We have added detailed explanations of the abbreviations of the three instruments in lines 77-79. In addition, we provide detailed descriptions of the three instruments in Section 1.1 of the supplementary.

Lines 88-89: "The detailed operations of AE33 and HR-ToF-AMS, and the data analysis are given in Xu et al. (in preparation).", some descriptions are needed because the paper of Xu et al. is in preparation.

Thank the reviewer for pointing this out. The dried ambient particles were collected by HR-ToF-AMS and AE33 at flow rates of 1 and 5 L min$^{-1}$ through stainless steel sampling lines (1/4 inch o. d.), respectively, and measured at time resolutions of 1 min in this study. Where the measurements of HR-ToF-AMS were performed under V-mode. The ionization efficiency (IE) was calibrated with ammonium nitrate particles (300 nm) and the elemental ratios of organic aerosols (OA) were

calculated with the "Improved-Ambient" method (Canagaratna et al., 2015; Jimenez et al., 2003; Jayne et al., 2000). The HR-ToF-AMS data were analyzed by using PIKA v 1.24, which showed that $NO_3$ (4.30 μg m$^{-3}$) and Org (3.80 μg m$^{-3}$) contributed 68.0 % of the mass concentration of NR-PM$_1$ (11.92 μg m$^{-3}$), followed by $SO_4$ (1.91 μg m$^{-3}$), $NH_4$ (1.69 μg m$^{-3}$), and Chl (0.22 μg m$^{-3}$). Thereafter, the sources of OA factors were resolved by using the positive matrix factorization (PMF) of high-resolution mass spectra of OA. Factors including fossil fuel combustion-related OA (FFBBOA), cooking OA (COA), and three SOA factors, i.e., two oxygenated OA (OOA1 and OOA2) and an aqueous-phase OOA were identified with mass concentrations of 0.31, 0.87, 0.83, 1.18 and 0.56 μg m$^{-3}$, respectively. In addition, the mass concentration of equivalent black carbon (eBC) obtained by AE33 was calculated based on the dual-spot measurement (Drinovec et al., 2015; Rajesh and Ramachandran, 2018), with an average of 1.34 μg m$^{-3}$ over the campaign. We have provided additional descriptions of the data analysis and operations of AE33 and HR-ToF-AMS in lines 89-97 as suggested. In addition, we have given the exact measurement values of AE33 and HR-ToF-AMS in the conclusion section when it is necessary to use them for specific demonstrations, e.g., lines 129 and 136, and Table 2 and S1.

Line 106: what are the sources of the two approaches of calculating the ρeff in this study? Besides, why do the authors use two approaches to calculate the ρeff?

Thank the reviewer's comments. As mentioned in the Response to Reviewer #1 above, we initially planned to connect DMA and AAC in tandem with SPAMS at different periods and select particles with $D_m$ and $D_a$ in the range of 150–300 nm and 200–700 nm, respectively, to finally obtain two complete datasets. However, only the SPAMS data with $D_a$ = 300 nm were eventually credible in the AAC-SPAMS period, accounting for 13.3% of the total particles captured by SPAMS (322415 of 2416964). This was due to the unstable sheath flow of AAC when selecting particles in the size range of 400–700 nm, and only 1756 particles were captured at $D_a$ = 200 nm due to the SPAMS detection limit. Considering that the data quality of the AAC-SPAMS period was unsatisfactory, we decided to combine the DMA-SPAMS and AAC-SPAMS data for the purpose of analyzing the data in order to ensure that the results of our study are representative. SPAMS, DMA and AAC can provide vacuum aerodynamic diameter ($D_{va}$), mobility diameters ($D_m$) and aerodynamic diameters ($D_a$) of particles, respectively. The relationship between these three diameters and the method for calculating the effective density ($\rho_{eff}$) of particles, which is obtainable when any two of these diameters are known, have been described in detail by Decarlo et al. (2004). For example, when the $D_{va}$ and $D_m$ of particles are available, the $\rho_{eff}$ of particles can be calculated as follows:

$$\rho_{eff} = \frac{D_{va}}{D_m} \rho_0 \tag{12}$$

This equation can be adopted to calculate the $\rho_{eff}$ of particles captured by the DMA-SPAMS tandem system, where $\rho_0$ is the standard density (1.0 g cm$^{-3}$). For the AAC-SPAMS tandem system, the $\rho_{eff}$ is defined based on the ratio of the particle density ($\rho_p$) to the particle dynamic shape factor ($\chi_\gamma$) as shown below:

$$\rho_{eff} = \frac{\rho_p}{\chi_\gamma} = \frac{D_{va}}{D_{ve}\rho_0} \tag{13}$$

The relationship between $D_a$, $D_{va}$ and $D_{ve}$ can be expressed by the following equation:

$$D_a = D_{ve}\sqrt{\frac{\rho_p C_c(D_{ve})}{\chi_t \rho_0 C_c(D_a)}} \tag{14}$$

where $\chi_t$ represents the aerosol dynamic shape factor in the transition regime. Considering the approximation between $\chi_t$ and $\chi_\gamma$, the $D_{ve}$ can be calculated by combining Eqs. (13) and (14) as follows:

$$C_c(D_a)\frac{D_a^2}{D_{va}} = D_{ve}C_C(D_{ve}) \tag{15}$$

$C_c(D)$ is the Cunningham slip correction factor, which can be calculated by the following equation:

$$Cc(D) = 1 + \frac{\lambda}{D}(A + B \cdot exp(\frac{C \cdot D}{\lambda})) \tag{16}$$

where $\lambda$ represents the mean free path of the gas molecules. A, B and C are empirically determined constants specific to the analyzed system, where A is 2.33, B is 0.966 and C is -0.498. Substituting Eq. (16) into Eq. (15) obtains Eq. (17):

$$\frac{D_a^2}{D_{va}} + \frac{D_a \cdot \lambda}{D_{va}}\left(A + B \cdot exp\left(\frac{C \cdot D_a}{\lambda}\right)\right) = D_{ve} + \lambda\left(A + B \cdot exp\left(\frac{C \cdot D_{ve}}{\lambda}\right)\right) \tag{17}$$

The $D_a$ and $D_{va}$ are known in the AAC-SPAMS tandem system, which can be brought into Eq. (17) to obtain $D_{ve}$. Finally, the $\rho_{eff}$ of particles captured by the AAC-SPAMS tandem system can be derived from Eq. (13). The accuracy of the above two methods of $\rho_{eff}$ calculation has been verified in previous studies (Spencer et al., 2007; Su et al., 2021; Peng et al., 2021; Pagels et al., 2009; Katrib et al., 2005). We have provided additional explanations in Section 2.3.2 of the manuscript as suggested.

Lines 498-502: the meaning of wind direction (WD) values should be described; the colors in curves of eBC and NR-PM1 cannot be well distinguished; "hit rare" is "hit rate"? Why "both size and hit counts after 2.10 are divided by 4"? Additionally, all the meaning of symbols (e.g., green cross in Fig. 1a) and lines (e.g., blue and green lines with arrows in Fig. 1a) occurred in figures should be described.

Thank the reviewer's comments. Wind direction is measured in degrees clockwise from due north (e.g. 0 degrees for a northerly wind and 270 degrees for a westerly wind) and has been explained on line 542 as suggested. In addition, the NR-PM$_1$ curve has been changed to red in the figure to distinguish it better. Line 543 was originally intended to express "hit rate", and we thank the reviewer for pointing out this spelling error. Only particles with $D_a$ of 300 nm were selected for data analysis in the AAC-SPAMS period, which is indicated by a blue arrow at the top of Fig. 2. While particles with $D_m$ of 200, 250, 300, and 150 nm were sequentially selected in the DMA-SPAMS period, indicated by four green arrows at the top of Fig. 2. The yellow and gray shading in the figure

represent the snowfall period and the Olympic Winter Games period, respectively. The meaning of the above elements in the figure has been further explained in lines 546-547.

As described in Section 2.1 of the manuscript, SPAMS was connected in tandem with DMA (21 January to 10 February) and AAC (10 February to 1 March) at different periods. Since the measurements of the DMA are performed by a force balance between the electrical force of a constant electric field on the net charges on the particle and the drag force experienced by the particle. Whereas the AAC selects particles based on aerodynamic sizes according to particle relaxation time without needing charging for electrostatic. In other words, DMA has more stringent conditions for particle selection than AAC, and more particles will be captured by SPAMS through AAC under the same circumstances. This is consistent with the results of this study, with average size and hit counts per minute of 49 and 16 when SPAMS was in tandem with DMA, compared to 396 and 84 when SPAMS was in tandem with AAC. The SPAMS hit rate was slightly different between the two periods (32.65% vs. 21.21%), which is attributed to the decrease in the hit rate due to the high pass rate of the AAC. However, if the time series of size and hit counts were plotted directly as shown in Fig. R4, the trend in the DMA-SPAMS period would be quite insignificant because too few particles were captured per minute compared to the AAC-SPAMS period. Therefore, in order to show the temporal trends of size and hit counts during the DMA-SPAMS period more visually, we reduced the data after 10 February to one-fourth of the original values. Additional explanations have been provided in lines 543-544 as suggested.

[Figure]

Figure R4: Time series of the number of sized particles, hit particles as well as the averaged hit rate of SPAMS per minute.

Line 158: "diurnal cycle" could be more reasonable than "diurnal trend".

Changed as suggested.

Line 254-255: Why did the emission controls during OWG lead to the increases in aged and regional particles? A reason analysis is expected in discussions. Thus, corresponding to the data analysis results, an analysis of mechanism behind the phenomenon is needed.

Thank the reviewer's comments. To improve air quality during the OWG period, the government took radical actions to reduce emissions from major sources (including industry, coal combustion, and transportation, among others), which resulted in a significant decrease in the number concentrations of particles captured during the OWG (Table S2). The number concentration of

Total-EC particles decreased by 61.80% during the OWG, which was much higher than 20.28% for Total-ECOC and 28.74% for Total-OC. Although emission control led to the overall decrease in the number concentration of particles during the OWG period, the role of aging and regional particles became more prominent. Especially for ECOC-NS particles, the particle counts of daily captures increased by 17.34% during the OWG period compared to the nOWG period. In addition, bivariate polar plots for most classes of particles show that the high number concentrations were concentrated at the sampling site during the nOWG period, while the source locations were skewed to the southeast or southwest during the OWG period. The $PA_{sulfate}/PA_{nitrate}$ of primary particles (e.g., KOC-N, HM, Biomass-K, etc.), which were determined by combining mass spectra, daily trends, and correlations with OA factors, were higher during the OWG period than during the nOWG period. This indicates that emission controls significantly reduce local primary emitted particles, resulting in significant aging and regional characteristics despite the small number concentrations of particles captured. We added Table S2 to the supplementary with the purpose of comparing the daily capture of different classes of particles during the OWG and nOWG periods. Additional explanations are also provided in lines 139-143.

Table S3: A summary of particle types and number of particles captured per day for the OWG and nOWG periods.

| Classification of particles | | OWG | nOWG | OWG (per day) | nOWG (per day) |
|---|---|---|---|---|---|
| Total-EC | pure-EC | 1155 | 2162 | 68 | 94 |
| | EC-NS | 8845 | 39761 | 520 | 1729 |
| | KEC-N | 4947 | 21303 | 291 | 926 |
| | KNaEC-N | 7039 | 14037 | 414 | 610 |
| Total-ECOC | ECOC-NS | 50382 | 58108 | 2964 | 2526 |
| | KECOC-CN | 3602 | 4891 | 212 | 213 |
| | KECOC-NS | 60631 | 109959 | 3567 | 4781 |
| | KNaECOC-NS | 13725 | 22096 | 807 | 961 |
| | KAECOC-NS | 662 | 23314 | 39 | 1014 |
| Total-OC | KOC-N | 13078 | 25298 | 769 | 1100 |
| | KOC-NS | 12242 | 18998 | 720 | 826 |
| | K-Amine-NS | 1219 | 6446 | 72 | 280 |
| Total-IA | K-N | 11561 | 24385 | 680 | 1060 |
| | KNa-N | 6215 | 9693 | 366 | 421 |
| Biomass-K | | 41627 | 54526 | 2449 | 2371 |
| HOM | | 16610 | 24388 | 977 | 1060 |
| Metals | rich-Fe | 1474 | 14088 | 87 | 613 |
| | other | 3542 | 7245 | 208 | 315 |

A discussion on the uncertainties of data quality and analysis should be included in the "conclusions" section.

Thanks to the reviewer for pointing this out. Although the real-time on-line measurement of the size and chemical composition of individual particles can be achieved by SPAMS, the ionizing laser has different sensitivities for the detection of different chemical compositions. For example, the ionizing laser is sensitive to alkaline metals (e.g., potassium and sodium) and elemental carbon particles with strong light absorption, leading to differences in the quantification of different chemical compositions. It is necessary to pay more attention to the evaluation of quantitative analysis of

SPAMS in the future. We have added discussion of instrumental uncertainty in lines 285-287 as suggested.

**References**

Canagaratna, M. R., Jimenez, J. L., Kroll, J. H., Chen, Q., Kessler, S. H., Massoli, P., Hildebrandt Ruiz, L., Fortner, E., Williams, L. R., Wilson, K. R., Surratt, J. D., Donahue, N. M., Jayne, J. T., and Worsnop, D. R.: Elemental ratio measurements of organic compounds using aerosol mass spectrometry: characterization, improved calibration, and implications, Atmos. Chem. Phys., 15, 253–272, doi:10.5194/acpd-14-19791-2014, 2015.

China, S., Salvadori, N., and Mazzoleni, C.: Effect of traffic and driving characteristics on morphology of atmospheric soot particles at freeway on-ramps, Environ. Sci. Technol., 48, 3128-3135, doi:10.1021/es405178n, 2014.

DeCarlo, P. F., Slowik, J. G., Worsnop, D. R., Davidovits, P., and Jimenez, J. L.: Particle Morphology and Density Characterization by Combined Mobility and Aerodynamic Diameter Measurements. Part 1: Theory, Aerosol Sci. Tech., 38, 1185-1205, doi:10.1080/027868290903907, 2004.

Drinovec, L., Močnik, G., Zotter, P., Prévôt, A. S. H., Ruckstuhl, C., Coz, E., Rupakheti, M., Sciare, J., Müller, T., Wiedensohler, A., and Hansen, A. D. A.: The "dual-spot" Aethalometer: an improved measurement of aerosol black carbon with real-time loading compensation, Atmos. Meas. Tech., 8, 1965-1979, doi:10.5194/amt-8-1965-2015, 2015.

Giorio, C., Tapparo, A., Dall'Osto, M., Beddows, D. C., Esser-Gietl, J. K., Healy, R. M., and Harrison, R. M.: Local and regional components of aerosol in a heavily trafficked street canyon in central London derived from PMF and cluster analysis of single-particle ATOFMS spectra, Environ. Sci. Technol., 49, 3330-3340, doi:10.1021/es506249z, 2015.

Hand, J. L. and Kreidenweis, S. M.: A New Method for Retrieving Particle Refractive Index and Effective Density from Aerosol Size Distribution Data, Aerosol Sci. Tech., 36, 1012-1026, doi:10.1080/02786820290092276, 2002.

Hand, J. L., Kreidenweis, S. M., Kreisberg, N., Hering, S., Stolzenburg, M., Dick, W., and McMurry, P. H.: Comparisons of Aerosol Properties Measured by Impactors and Light Scattering from Individual Particles: Refractive Index, Number and V olume Concentrations, and Size Distributions, Atmos. Environ., 36, 1853–1861, doi:10.1016/S1352-2310(02)00103-6, 2002.

Hu, J., Xie, C., Xu, L., Qi, X., Zhu, S., Zhu, H., Dong, J., Cheng, P., and Zhou, Z.: Direct Analysis of Soil Composition for Source Apportionment by Laser Ablation Single-Particle Aerosol Mass Spectrometry, Environ. Sci. Technol., 55, 9721-9729, doi:10.1021/acs.est.0c07983, 2021.

Jayne, J. T., Leard, D. C., Zhang, X., Davidovits, P., Smith, K. A., Kolb, C. E., and Worsnop, D. R.: Development of an aerosol mass spectrometer for size and composition analysis of submicron particles, Aerosol Sci. Technol., 33, 49–70, 2000.

Jimenez, J. L.: New particle formation from photooxidation of diiodomethane (CH2I2), J. Geophys. Res., 108, doi:10.1029/2002jd002452, 2003.

Jimenez, J. L., Jayne, J. T., Shi, Q., Kolb, C. E., Worsnop, D. R., Yourshaw, I., Seinfeld, J. H., Flagan, R.

C., Zhang, X., Smith, K. A., Morris, J. W., and Davidovits, P.: Ambient aerosol sampling using the Aerodyne Aerosol Mass Spectrometer, J. Geophys. Res. , 108, 8425, doi:10.1029/2001jd001213, 2003.

Katrib, Y., Martin, S. T., Rudich, Y., Davidovits, P., Jayne, J. T., and Worsnop, D. R.: Density changes of aerosol particles as a result of chemical reaction, Atmos. Chem. Phys., 5, 275–291, doi:10.5194/acp-5-275-2005, 2005.

Liu, H., Pan, X., Wu, Y., Wang, D., Tian, Y., Liu, X., Lei, L., Sun, Y., Fu, P., and Wang, Z.: Effective densities of soot particles and their relationships with the mixing state at an urban site in the Beijing megacity in the winter of 2018, Atmos. Chem. Phys., 19, 14791-14804, doi:10.5194/acp-19-14791-2019, 2019.

McMurry, P. H., Wang, X., Park, K., and Ehara, K.: The Relationship between Mass and Mobility for Atmospheric Particles: A New Technique for Measuring Particle Density, Aerosol Sci. Technol., 36, 227-238, doi:10.1080/027868202753504083, 2002.

Pagels, J., Khalizov, A. F., McMurry, P. H., and Zhang, R. Y.: Processing of Soot by Controlled Sulphuric Acid and Water Condensation—Mass and Mobility Relationship, Aerosol Sci. Technol., 43, 629-640, doi:10.1080/02786820902810685, 2009.

Peng, L., Li, L., Zhang, G., Du, X., Wang, X., Peng, P. a., Sheng, G., and Bi, X.: Technical note: Measurement of chemically resolved volume equivalent diameter and effective density of particles by AAC-SPAMS, Atmos. Chem. Phys., 21, 5605-5613, doi:10.5194/acp-21-5605-2021, 2021.

Rajesh, T. A. and Ramachandran, S.: Black carbon aerosol mass concentration, absorption and single scattering albedo from single and dual spot aethalometers: Radiative implications, J. Aerosol Sci., 119, 77-90, doi:10.1016/j.jaerosci.2018.02.001, 2018.

Sodeman, D. A., Toner, S. M., and Prather, K. A.: Determination of Single Particle Mass Spectral Signatures from Light-Duty Vehicle Emissions, Environ. Sci. Technol. , 39, 4569-4580, doi:10.1021/es0489947 2005.

Spencer, M. T., Shields, L. G., and Prather, K. A.: Simultaneous Measurement of the Effective Density and Chemical Composition of Ambient Aerosol Particles, Environ. Sci. Technol., 41, 1303-1309, doi:10.1021/es061425+ 2007.

Su, B., Zhang, G., Zhuo, Z., Xie, Q., Du, X., Fu, Y., Wu, S., Huang, F., Bi, X., Li, X., Li, L., and Zhou, Z.: Different characteristics of individual particles from light-duty diesel vehicle at the launching and idling state by AAC-SPAMS, J. Hazard. Mater., 418, 126304, doi:10.1016/j.jhazmat.2021.126304, 2021.

Wang, X., Ye, X., Chen, J., Wang, X., Yang, X., Fu, T.-M., Zhu, L., and Liu, C.: Direct links between hygroscopicity and mixing state of ambient aerosols: estimating particle hygroscopicity from their single-particle mass spectra, Atmos. Chem. Phys., 20, 6273-6290, doi:10.5194/acp-20-6273-2020, 2020.

Zhang, G., Bi, X., Qiu, N., Han, B., Lin, Q., Peng, L., Chen, D., Wang, X., Peng, P. a., Sheng, G., and Zhou, Z.: The real part of the refractive indices and effective densities for chemically segregated ambient aerosols in Guangzhou measured by a single-particle aerosol mass spectrometer, Atmospheric Chemistry

and Physics, 16, 2631-2640, doi:10.5194/acp-16-2631-2016, 2016.

Zhang, Y., Pei, C., Zhang, J., Cheng, C., Lian, X., Chen, M., Huang, B., Fu, Z., Zhou, Z., and Li, M.: Detection of polycyclic aromatic hydrocarbons using a high performance-single particle aerosol mass spectrometer, J. Environ. Sci., 124, 806-822, doi:10.1016/j.jes.2022.02.003, 2022.